# DeepRePath: Identifying the Prognostic Features of Early-Stage Lung Adenocarcinoma Using Multi-Scale Pathology Images and Deep Convolutional Neural Networks

**DOI:** 10.3390/cancers13133308

**Published:** 2021-07-01

**Authors:** Won Sang Shim, Kwangil Yim, Tae-Jung Kim, Yeoun Eun Sung, Gyeongyun Lee, Ji Hyung Hong, Sang Hoon Chun, Seoree Kim, Ho Jung An, Sae Jung Na, Jae Jun Kim, Mi Hyoung Moon, Seok Whan Moon, Sungsoo Park, Soon Auck Hong, Yoon Ho Ko

**Affiliations:** 1Deargen Inc., Daejeon 34051, Korea; wsshim@deargen.me (W.S.S.); sspark@deargen.me (S.P.); 2Department of Hospital Pathology, College of Medicine, The Catholic University of Korea, Seoul 06591, Korea; kangse_manse@catholic.ac.kr (K.Y.); kimecho@catholic.ac.kr (T.-J.K.); yesung@catholic.ac.kr (Y.E.S.); 3Department of Pathology, College of Medicine, Chung-Ang University, Seoul 06974, Korea; gyeongyooni@gmail.com; 4Division of Oncology, Department of Internal Medicine, College of Medicine, The Catholic University of Korea, Seoul 06591, Korea; jh_hong@catholic.ac.kr (J.H.H.); rowett@naver.com (S.H.C.); seoreek@gamil.com (S.K.); meicy@catholic.ac.kr (H.J.A.); 5Department of Radiology, Uijeongbu St. Mary’s Hospital, College of Medicine, The Catholic University of Korea, Seoul 06591, Korea; sj0405@catholic.ac.kr; 6Department of Thoracic and Cardiovascular Surgery, College of Medicine, The Catholic University of Korea, Seoul 06591, Korea; medkjj@hanmail.net (J.J.K.); sophiamoon@catholic.ac.kr (M.H.M.); swmoon@catholic.ac.kr (S.W.M.); 7Cancer Research Institute, College of Medicine, The Catholic University of Korea, Seoul 06591, Korea

**Keywords:** deep learning, lung adenocarcinoma, pathology image, prognosis

## Abstract

**Simple Summary:**

Pathology images are vital for understanding solid cancers. In this study, we created DeepRePath using multi-scale pathology images with two-channel deep learning to predict the prognosis of patients with early-stage lung adenocarcinoma (LUAD). DeepRePath demonstrated that it could predict the recurrence of early-stage LUAD with average area under the curve scores of 0.77 and 0.76 in cohort I and cohort II (external validation set), respectively. Pathological features found to be associated with a high probability of recurrence included tumor necrosis, discohesive tumor cells, and atypical nuclei. In conclusion, DeepRePath can improve the treatment modality for patients with early-stage LUAD through recurrence prediction.

**Abstract:**

The prognosis of patients with lung adenocarcinoma (LUAD), especially early-stage LUAD, is dependent on clinicopathological features. However, its predictive utility is limited. In this study, we developed and trained a DeepRePath model based on a deep convolutional neural network (CNN) using multi-scale pathology images to predict the prognosis of patients with early-stage LUAD. DeepRePath was pre-trained with 1067 hematoxylin and eosin-stained whole-slide images of LUAD from the Cancer Genome Atlas. DeepRePath was further trained and validated using two separate CNNs and multi-scale pathology images of 393 resected lung cancer specimens from patients with stage I and II LUAD. Of the 393 patients, 95 patients developed recurrence after surgical resection. The DeepRePath model showed average area under the curve (AUC) scores of 0.77 and 0.76 in cohort I and cohort II (external validation set), respectively. Owing to low performance, DeepRePath cannot be used as an automated tool in a clinical setting. When gradient-weighted class activation mapping was used, DeepRePath indicated the association between atypical nuclei, discohesive tumor cells, and tumor necrosis in pathology images showing recurrence. Despite the limitations associated with a relatively small number of patients, the DeepRePath model based on CNNs with transfer learning could predict recurrence after the curative resection of early-stage LUAD using multi-scale pathology images.

## 1. Introduction

Among all cancer types, lung cancer is the leading cause of cancer-related deaths, accounting for 25% of cancer-related deaths [1]. Non-small cell lung cancer (NSCLC) is the most common type of lung cancer, and lung adenocarcinoma (LUAD) accounts for more than 50% of all cases of NSCLC. Recently, the clinical outcomes of LUAD patients have been greatly improved with the development of effective treatment approaches, including surgical or radiation techniques, and the introduction of targeted therapies and immunotherapies tailored to the molecular or immunological characteristics of primary tumors [2]. However, the survival rate for curatively resected LUAD remains low, ranging from 58% to 73% in stage I, 36% to 46% in stage II, and only 19% to 24% in stage IIIA [3]. To achieve better clinical outcomes, adjuvant chemotherapy is required for resected NSCLC; however, questions remain as to which patients benefit from adjuvant chemotherapy. Therefore, the accurate and timely identification of patients with a high risk of recurrence may provide opportunities to optimize clinical interventions for patients with early-stage LUAD.

Similar to that of cancers, the diagnosis of LUAD is completely dependent on histopathological findings. In particular, the histopathological features of LUAD provide crucial information related to prognosis as well as diagnosis [4,5]. The pathologic subtypes of LUAD (lepidic, acinar, papillary, solid, and micropapillary) can contribute to different recurrence and survival rates [6,7,8,9,10]. New histomorphological features of LUAD have been identified with increasing frequency, including tumor budding, lymphovascular invasion, and tumor spread through air spaces [11,12,13,14]. Although microscopic morphology has predictive and prognostic values, the interpretation of pathology images is time-consuming and error-prone for pathologists and subjected to interobserver and intraobserver variabilities [15,16]. Recently, deep learning using histopathology images has emerged as a new tool to aid pathologists in various tasks in clinical settings. In comparison with human pathologists, artificial intelligence using deep learning and pathology images has advantages, such as improved reproducibility and consistency in recognizing diagnostic clues and pathological patterns, as well as continuous estimation of the immunolabeling index and cell count [17]. Notably, one of the benefits of using artificial intelligence in pathology is the identification of novel features, that is, subvisual features that could be helpful for supporting decisions in patient management [17]. Several related studies on predicting the prognosis of patients with NSCLC have been published [18,19,20,21,22,23,24,25,26,27]. Previous studies used deep learning to evaluate prognosis-associated histopathological variables, such as tumor-infiltrating lymphocytes, necrosis, tumor stroma, and nuclear segmentation [19,20]. In these studies, deep learning was usually only used as part of the process of evaluating histopathological variables. Features such as tumor regions or nuclei were detected using deep learning, and dozens to hundreds of morphological features were extracted from the information obtained using deep learning to evaluate histopathological variables using several geometrical methods. The prognosis of patients with NSCLC was predicted with some degree of success on the basis of these morphological features, using existing machine learning methods such as support vector machine (SVM) and Cox proportional hazards (CoxPH) [19,20]. However, the process of extracting a large number of morphological features from images and selecting important features from them is complicated and cumbersome.

Therefore, in this study, we predicted recurrence directly with the histopathology images of LUAD, using only deep learning without extracting predefined morphological features (DeepRePath). We aimed to demonstrate that DeepRePath could predict the prognosis of patients with early-stage LUAD, which may facilitate treatment decision-making.

## 2. Materials and Methods

### 2.1. Study Population and Baseline Characteristics

The clinical and pathological data of NSCLC patients who had undergone curative resection between 2009 and 2017 at five St. Mary’s hospitals affiliated with the Catholic University of Korea in Seoul, Incheon, Uijeongbu, Bucheon, and Yeouido were reviewed. The inclusion criteria were as follows: (i) pathologically confirmed stage I–II LUAD; (ii) availability of a pathology report; (iii) no preoperative radiation or chemotherapy; and (iv) at least 3 years of follow-up. The clinicopathological characteristics of cohort I and cohort II (external validation set) are summarized in Appendix A. A total of 1104 patients underwent lung cancer surgery between 2009 and 2017, and of these, 393 patients who met the inclusion criteria were selected from the six hospitals. Of these patients, 302 patients were included in the training, validation, and test sets (cohort I), and the remaining 91 patients were included in the external validation set (cohort II) (Appendix A).

The median age of the patients was 64 years (range = 25–86 years), 50.1% of the patients were men, and 79.9% of the patients were classified as stage I. The baseline characteristics were not significantly different between cohort I and cohort II, except for sex and lymphovascular invasion. The median follow-up periods of cohort I and cohort II were 59.9 (range = 6.7–99.2) and 60.4 (range = 12.2–108.6) months, respectively, and 72 (23.8%) and 22 (24.2%) patients experienced recurrence at 3 years after curative resection, respectively. This study was approved by the institutional review board of the Catholic Medical Center (no. UC17SESI0073) and was performed in accordance with the guidelines of human research. The requirement for written informed consent was waived by the institutional review board (Catholic Medical Center) because of the retrospective nature of this study.

### 2.2. Data Preparation

A total of 3923 hematoxylin and eosin (HE)-stained slides were collected form 393 patients. Three board-certified pathologists reviewed all the slides and selected one representative slide for each case. Representative slides at 40×, 100×, 200× and 400× magnification were captured by three pathologists (S.A.H., K.Y., and T.-J.K.). In total, 5 pathological images (40×, 2 images; 100×, 1; 200×, 1; 400×, 1 image) were obtained from each case using Olympus DP74 (Olympus, Tokyo, Japan). Tumors with adjacent non-neoplastic tissues were captured in the 40× images. In the 100× tumor images, the pathologists focused on and captured the prevalent and aggressive architecture of the tumors. Tumor cells with aggressive cytological features, including nuclear hyperchromasia, pleomorphism, membrane irregularity, and high nuclear-cytoplasmic ratio, were captured in 200× and 400× magnification. All images were re-examined and confirmed by two pathologists (S.A.H. and K.Y.) (Appendix A). The 100× and 400× tumor images were finally selected as the input image data because DeepRepath showed the best performance (higher value of area under the curve (AUC)) with a combination of 100× and 400× tumor images.

A total of 302 patients were included in cohort I (training and validation sets). The patients in cohort I were randomized to maintain the ratio of the training (74%), validation (8%), and test (18%) sets. However, the training classes were unbalanced. LUAD recurred within 3 years in only 24% of patients in cohort I. To resolve the class imbalance problem, we used image data augmentation techniques, such as vertical flip, horizontal flip, and standardization. We oversampled recurrence cases and increased them to 50% in the training set. The training and validation sets were used for model learning and optimal model selection, and the test set was used to evaluate the performance of the model. In addition, 91 cases (182 images with 5450 patches) were used in cohort II (external validation set). Data in cohort I were obtained from three St. Mary’s hospitals affiliated with the Catholic University of Korea in Seoul, Incheon, and Uijeongbu. Data in cohort II (external validation set) were obtained from two St. Mary’s hospitals affiliated with the Catholic University of Korea in Bucheon and Yeouido.

### 2.3. Deep Learning-Based Recurrence Prediction Using Histopathology Images of LUAD in the DeepRePath Model

#### 2.3.1. Pre-Training for Transfer Learning

Training a deep learning model with insufficient data is highly likely to cause the model to be overfitted. When training data are insufficient, transfer learning using a pre-trained model is a common method that is employed to prevent overfitting. As we had insufficient histopathology image data to train the model for predicting the prognosis of lung cancer patients, we performed transfer learning using data from a similar domain. We obtained 1067 (tumor = 823, normal = 244) HE-stained whole-slide histopathology images (WSIs) of LUAD from the Cancer Genome Atlas (TCGA). We pre-trained a convolutional neural network (CNN) using TCGA data to classify LUAD and normal images. Figure 1A shows the pre-training workflow. We extracted the 10 core tiles, excluding the relatively white areas in the histopathology image. The core tiles were captured at 40× magnification, and the size was 1024 × 1024 pixels. We extracted 20 patches from each tile and trained the CNN (ResNet50) using these patches as inputs. We then fine-tuned this pre-trained model using histopathology image data of 302 patients from St. Mary’s hospitals.

#### 2.3.2. Model Architecture

Both tumor cell patterns and structural patterns are critical factors in predicting recurrence. As the two patterns have different characteristics, it is not effective to perform training simultaneously with one network. Therefore, we constructed DeepRePath for multi-scale pathology images using two separate CNNs (ResNet50). Figure 1B shows the architecture of DeepRePath. One network was used for structural patterns, and its input included the images captured at 100× magnification. The other network was for tumor cell patterns, and its input included the images captured at 400× magnification. We concatenated the two feature vectors extracted from the two networks and performed classification using XGBoost. Figure 1C shows the DeepRePath classification workflow. The images were augmented using adjustments of vertical flip, horizontal flip, and standardization, to fix the class imbalance caused by the low proportion of cases with recurrence. In the case of a 400× magnification image, 36 patches with a size of 224 × 224 pixels were extracted from one image of tumor cell patterns. In the case of a 100× magnification image, DeepRePath extracted 24 patches with a size of 500 × 500 pixels and then reduced them to a size of 224 × 224 pixels. Patches extracted from a 100× magnification image and patches extracted from a 400× magnification image were each passed through the CNN independently. The CNN extracted feature vectors with a size of 2048 from each patch, and the feature vectors were averaged element by element. The two averaged feature vectors from both the 400× and 100× magnification images were concatenated. Finally, we predicted the probability of recurrence within 3 years using XGBoost.

#### 2.3.3. Visualization of the DeepRePath Model

To create a deep learning application that analyzes histopathology images, it is crucial to make the deep learning model more interpretative so that pathologists can understand it. To visualize our CNN-based DeepRePath model, we used the gradient-weighted class activation mapping (Grad-CAM) algorithm, which provides an explainable heatmap of the CNN model [28]. The input image was forward-propagated through the CNN of the DeepRePath to obtain the raw score of the recurrence. Then, Grad-CAM back-propagated this signal to the convolutional layer of interest so as to obtain the gradient. Using both these values, Grad-CAM computed the convolutional feature maps and combined them to compute the heatmap.

In addition to heatmap visualization analysis, morphometric analysis of the nucleus was performed using ImageJ software (National Institutes of Health, Bethesda, MD, USA) [29,30]. We selected 30 representative patches at a magnification of 400×. In each patch, all nuclei in the hotspots and the same number of coldspot nuclei were analyzed. After drawing along the nucleus, we obtained the area, primary and secondary axes, maximum and minimum Feret diameters, and the perimeter to evaluate the nuclear size and length. Moreover, the shape factor and roughness were determined to evaluate nuclear irregularity, and the aspect ratio and roundness were determined to evaluate nuclear elongation.

### 2.4. Statistical Analysis

Disease-free survival (DFS) duration was defined as the time from the date of surgery until the first recurrence or death from any cause, whichever was observed first, and the survival curves were estimated using the Kaplan–Meier method and compared using the log–rank test. The performance of our models was measured and compared using AUC scores. To determine the AUC of the classification for 3-year recurrence, patients censored before 3 years were excluded from the test set because recurrence classification for these samples was unclear. The evaluation matrix included accuracy, sensitivity, specificity, positive predictive value, and negative predictive value. The nuclear morphometric results were evaluated using an independent *t*-test. Survival curves of the external cohort were generated using the Kaplan–Meier method and compared using the log–rank test. In the multivariate analysis, CoxPH regression models from the external cohort were used to identify the significance of the prognostic factors. Survival rates and hazard ratios are shown with their respective 95% confidence intervals (CIs). All statistical analyses were performed using R statistical programming (version 3.4.1; http://www.r-project.org), the SPSS software package (version 23, IBM, Chicago, IL, USA) and GraphPad Prism 8.0 (GraphPad Software, Inc., Graphpad Holdings, San Diego, CA, USA). A two-sided *p*-value of <0.05 was considered statistically significant in all the tests and models.

## 3. Results

### 3.1. Model Performance

Patients in cohort I were randomized to maintain the ratio of the training (74%), validation (8%), and test (18%) sets. The training and validation sets were used for model learning and optimal model selection, and the test set was used to evaluate model performance. A five-fold stratified cross-validation was used for training and validation.

The performance results of our DeepRePath model are presented in Table 1 and Figure 2. The model performance was evaluated by averaging the scores of the five-fold stratified cross-validation. In the DeepRePath model, the use of 100× magnification images showed a sensitivity of 65%, a specificity of 59%, an accuracy of 62%, and an AUC score of 0.6, while the use of 400× magnification images showed a sensitivity of 52%, a specificity of 78%, an accuracy of 71%, and an AUC score of 0.68. In contrast, the use of 100× and 400× magnification images showed the best single model performance with a sensitivity of 46%, a specificity of 94%, an accuracy of 82%, and an AUC score of 0.72. These findings indicate that the features of the structural and tumor cell images were complementary to each other in deep learning for predicting clinical outcomes. After data augmentation, the model performance was improved to a sensitivity of 74%, a specificity of 78%, and an AUC score of 0.77, but the accuracy was decreased to 77%.

To further analyze the robustness, reproducibility, and reliability of the model, we performed an additional validation using data from cohort II. To develop the final DeepRePath model for testing cohort II, we used all the data from cohort I, including the five-fold cross-validation test set, to train the model. Similar to the results of cohort I, the use of architectural and tumor cell images showed the best single model performance compared with the use of only structural or tumor cell images (accuracy of 77% and AUC of 0.76; Table 2 and Figure 3). In addition, we compared the results with or without transfer learning. Table 3 presents the differences between the models with and without transfer learning. These models were trained without data augmentation. When transfer learning was not applied, the AUC score of cohort I was high at 0.87, while the AUC score of cohort II (external validation set) was very low (0.58) (Table 3). This result suggests overfitting. When transfer learning was applied, the AUC score of cohort I was slightly reduced to 0.72, while the AUC score of cohort II was relatively high at 0.75. Therefore, we could prevent overfitting and obtain a more generalized model through transfer learning. We also compared three boosting algorithms: XGBoost, Gradient Boosting, and Adaptive Boosting (AdaBoost). In the DeepRePath model, the use of XGB showed the best model performance compared to the use of Gradient Boosting or AdaBoost (Appendix A).

### 3.2. Visualization

To identify the area in the pathology image that is the most responsible for predicting recurrence in the DeepRePath model, Grad-CAM was used to visualize our CNN-based DeepRePath model by creating heatmaps (Figure 4). The image produced by Grad-CAM was examined and assessed by two pathologists (S.A.H and K.Y). Grad-CAM highlighted the atypical nuclei of tumor cells under a high-power view (400×), which contributed to recurrence (Figure 4A,B). Under a low-power view (100×) of the architectural patterns of the tumor, Grad-CAM revealed two main histological features that may be associated with a high probability of recurrence—the first was tumor necrosis (Figure 4C), and the second was discohesive tumor cells in the alveolar space (Figure 4D).

### 3.3. Nuclear Morphometric Results of Hotspots and Coldspots in Heatmap Visualization

To clarify the association between heatmap visualization and nuclear morphology, morphometric analysis of the nucleus was performed using ImageJ software. The results are summarized in Table 4. The nuclear size, primary and secondary axes, Feret diameters, and perimeter were significantly greater in hotspots than in coldspots (all *p* < 0.001), whereas the shape factor was significantly lower in hotspots than in coldspots (*p* = 0.036). These findings indicate that nuclear enlargement and membrane irregularity could predict recurrence, while nuclear elongation was not associated with recurrence.

### 3.4. Prognostic Significance of the DeepRePath Model

To determine the clinical significance of the DeepRePath model, survival analysis for DFS was performed on patients in the external validation set (cohort II). We assumed that the samples with a high probability score would have a high-risk probability; hence, we sorted the samples according to the probability scores based on the optimal probability threshold using the receiver operating characteristic curve. Patients with high-risk scores demonstrated significantly shorter DFS in each individual stage (I and II), as well as stages I and II together (total population, Figure 5A, *p* < 0.0001; stage I, Figure 5B, *p* = 0.0009; stage II, Figure 5C, *p* = 0.0005). Notably, patients with stage I disease are of special interest because postoperative treatment for stage I disease is controversial. In this study, despite the small number of patients with stage I disease, the DFS of patients with high-risk and low-risk scores in the DeepRePath model was significantly different (*p* = 0.0009, Figure 5). In the univariate analyses, patients with a high-risk score based on the DeepRePath model had statistically worse outcomes (Table 5; *p* < 0.001). In the multivariate CoxPH analysis, DeepRePath model scores remained a statistically significant predictor of recurrence with a high score, indicating an unfavorable prognosis (Table 5; hazard ratio = 5.564, 95% confidence interval = 2.245–13.789, *p* < 0.001).

## 4. Discussion

Although histopathology images provide clinicians with important information related to a patient’s clinical outcomes, it is challenging for pathologists to predict recurrence from most images after the curative resection of early-stage LUAD. In this study, we developed a deep learning model (DeepRePath) to predict the recurrence of primary tumors. DeepRePath demonstrated good performance using multi-scale pathology images that show the tumor architecture and tumor cells. Our DeepRePath model could stratify early-stage LUAD into high-risk and low-risk groups.

Previously, various algorithms using machine learning have been developed to analyze pathology images for cancer detection [31], assessment of the histologic growth pattern [32] and PD-L1 status [33], histological subtyping [34], microenvironment analysis [20,21,25,27], and nuclear segmentation [23]. The prognostic value of these analyses has been described in some studies [18,19,20,21,22,23,24,25,26,27]. The performance of our model (specificity = 78%, sensitivity = 74%, accuracy = 77%, hazards ratio = 5.564, and AUC score = 0.76–0.77) was superior or similar to that of the previously described models [18,19,20,21,22,23,24,25,26,27]. However, these studies first elucidated the specific values, including tumor shape and boundary features [19], tumor cell features [18,23,24], and tumor microenvironment [20,21,25,27], which have already been reported as potential prognostic factors. Thereafter, they attempted to predict prognosis based on these factors. Therefore, these studies were limited with respect to identifying novel prognostic factors. Conversely, we selected the specific tumor area and simply analyzed these images using artificial intelligence with minimal interference in predicting prognosis or recurrence.

Wang et al. tried to predict recurrence in early-stage NSCLC through nuclear segmentation and nuclear feature extraction using CellProfiler. They categorized the extracted features using three popular classifiers (QDA, LDA, and SVM with polynomial kernel) and predicted recurrence (AUC score = 0.69–0.84, accuracy = 75–82%) [23]. However, only 122 patients with LUAD were enrolled in that study, and the authors did not present the data regarding the performance of their model based on the histological type (adenocarcinoma vs. squamous cell carcinoma). In contrast to the previous study, we performed data augmentation to circumvent issues associated with the low incidence of early-stage lung adenocarcinoma recurrence [23]. Additionally, in our study, pathologists examined foci that were indicative of a high probability of recurrence, depicted them as heatmaps, and tried to interpret these features. As a result, nuclear atypia, tumor necrosis, and discohesive tumor cells were identified as the potential prognostic features.

Similarly, after nuclear segmentation and extraction with CellProfiler, Yu et al. used seven classifiers to differentiate malignant and normal tissues and adenocarcinoma from squamous cell carcinoma, and finally predicted the survival in stage 1 LUAD (Kaplan–Meier curve, *p* = 0.0023–0.028) [24]. In addition, the nuclei were analyzed using machine learning with nuclear feature extraction. Luo et al. selected significant features using CoxPH analysis and then predicted the survival using random forest methods (hazard ratio = 2.34 in LUAD and 2.22 in LUSC) [18]. However, the features identified from nuclear segmentation are difficult to utilize in clinical settings, and the removal of false-positives is not feasible. Therefore, we allowed the deep learning model to freely predict recurrence. The nuclei were mainly indicated as hotspots on the heatmap. Recurrence was associated with an enlarged nucleus and an irregular nuclear membrane, but not nuclear elongation (Table 4). Thus, we could eliminate potential false-positive findings and obtain results that are applicable to actual clinical practice.

We used pathology images with different scales to input the data for DeepRePath. Depending on the scale, the images have their own strengths for extracting pathologic features, such as tumor-infiltrating immune cells, tumor cells, and tumor stroma. In our study, we found that DeepRePath extracted tumor necrosis and tumor cell patterns under a low-power view (original magnification, 100×). Tumor necrosis and discohesive tumor cells may be key features related to aggressive tumor behavior [35,36]. However, architectural features related to prognoses, such as micropapillary features, lymphovascular invasion, increased stroma, and tumor spread through air spaces, could not be identified in our model. Further studies on integrating prognostic features identified by human pathologists and deep learning models could be helpful for accurately predicting prognosis.

Adjuvant chemotherapy plays a significant role in the treatment of patients with resected LUAD and improves overall survival, resulting in a 4–5% absolute increase in the 5-year survival [37]. Although adjuvant chemotherapy is limited to patients with above stage I NSCLC, in a previous study, 30% of patients in that stage showed disease recurrence [38]. In practice, clinicians experience difficulties in deciding on chemotherapy to prevent recurrence for patients with stage I disease. DeepRePath by itself might be limited with respect to detecting recurrence owing to its low performance. However, in combination with clinicopathological features (TNM stage, solid and micropapillary subtypes), DeepRePath can aid clinicians in diagnosing patients with recurrence who might benefit from adjuvant chemotherapy in early-stage lung adenocarcinoma. Additionally, DeepRePath may enable the determination of the duration of radiology examinations and follow-up periods for the early detection of recurrence.

In our study, predicting the recurrence of lung cancer by analyzing pathology images showed significant results; however, there were some limitations. First, the number of pathology images used for deep learning was relatively small in the context of developing a general model for predicting stage I and II LUAD. Further studies using more data are needed to determine whether DeepRePath can predict various LUAD stages. Second, in this study, pathologists identified a large section of tumor cells on histopathology slides and input the captured images into our deep learning model to determine lung cancer recurrence. To effectively train the model using WSIs without pathologist intervention, a fully automated model that can detect and segment the tumor part is needed. In a previous study, a model using WSIs and deep learning failed to predict disease-specific survival in the case of lung adenocarcinoma (hazard ratio = 1.35, confidence interval = 0.87–2.08, *p* = 0.1824) [22]. However, Wu et al. predicted the recurrence of lung cancer using WSIs from TCGA and deep convolutional neural networks. They reported a relatively good prediction performance (AUC score = 0.79, sensitivity = 0.84, and specificity = 0.67), which was similar to that of DeepRePath [26]. Nonetheless, WSIs require high computing power, and there are problems with image quality and confounding non-neoplastic tissues outside the tumor area [39]. Practically, the selection and refinement of images by experienced pathologists can reduce these disadvantages. All our data from the input pathology images were selected and refined by three pathologists. Third, to identify the area in the pathology image that is the most responsible for predicting recurrence in the DeepRePath model, we used Grad-CAM to visualize the CNN result by creating heatmaps. Nevertheless, some heatmaps were difficult to interpret for the pathologists. To determine whether specific heatmap regions are representative of novel histopathological features, more data should be collected to establish the consistency of the results. Further studies may be warranted based on these results.

## 5. Conclusions

In conclusion, our findings show that a DeepRePath model with transfer learning using two separate CNNs could identify early-stage LUAD patients with a high risk of recurrence based on multi-scale pathology images, despite some limitations related to the small number of patients. Although DeepRePath is not suitable for use as an automated tool in clinical settings owing to its low performance, differential risk classification using the DeepRePath model can facilitate patient prognosis. Ultimately, our results demonstrate the usefulness of a deep learning model to clinically stratify patients beyond the TNM stage. This contributes to the development of personalized treatments that can improve patient outcomes.

## Figures and Tables

**Figure 1 cancers-13-03308-f001:**
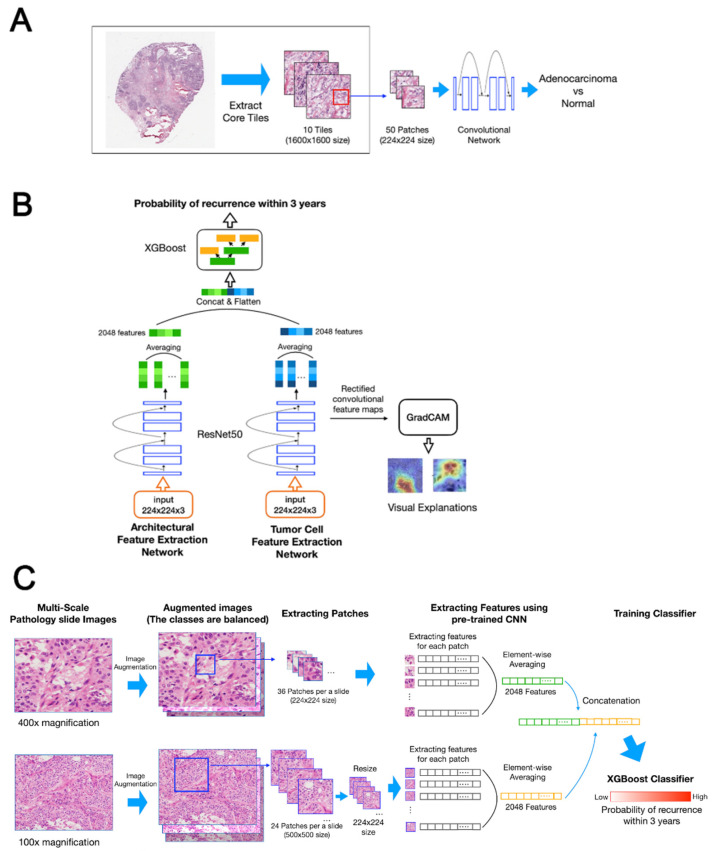
DeepRePath model. (**A**) Pre-training workflow. A total of 10 core tiles (1600 × 1600 pixels) excluding white areas were captured at 40× magnification from whole-slide histopathology images of lung adenocarcinoma (LUAD) from the Cancer Genome Atlas (TCGA). A total of 50 patches (224 × 224 pixels) were extracted from the core tiles. Resnet50 was trained to classify adenocarcinoma vs. normal tissue using the patches. This pre-trained Resnet50 model was used for transfer learning of the DeepRePath model. (**B**) DeepRePath network architecture. For multi-scale pathology images, DeepRePath was constructed using two separate convolutional neural networks (CNNs) (ResNet50). One network was for tumor cell patterns, and its input was the images captured at 400× magnification. The other network was for structural patterns, and its input was the images captured at 100× magnification. A total of 2048 features produced by both networks were concatenated into 4096 features. These features were used as inputs for the XGBoost classifier trained to predict the probability of recurrence within 3 years. For visualization of the DeepRePath model, the gradient-weighted class activation mapping (Grad-CAM) algorithm was used. (**C**) DeepRePath training workflow: Slide images whose class is in the minority are augmented by methods such as vertical flip, horizontal flip and standardization to fix class imbalance. In the case of a 400× magnification image, 36 patches with a size of 224 × 224 pixels were extracted from an image for tumor cell patterns. In the case of a 100× magnification image, the DeepRePath extracted 24 patches with a size of 500 × 500 pixels and then reduced them to a size of 224 × 224 pixels. Patches extracted from a 100× magnification image and patches extracted from a 400× magnification image were each passed through the CNN independently. The CNN extracted feature vectors with a size of 2048 from each patch, and the feature vectors were averaged element by element. The two averaged feature vectors from both 400× and 100× magnification images were concatenated. Finally, we predicted the probability of recurrence within 3 years using XGBoost.

**Figure 2 cancers-13-03308-f002:**
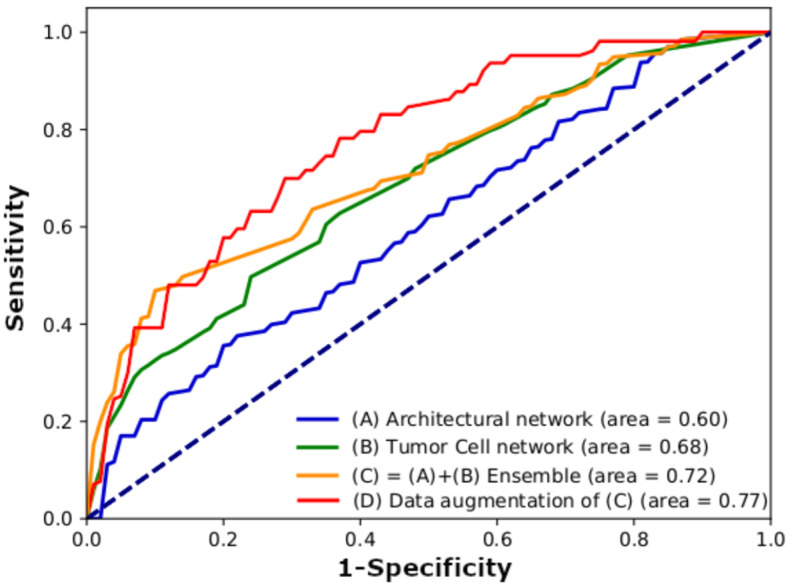
Comparison of the performance between mono-scale histopathology image processing network (**A**,**B**) and multi-scale histopathology image processing network (**C**,**D**) based on ROC area under the curve (AUC). (**A**) Architectural network (100× magnification). (**B**) Tumor cell network (400× magnification). (**C**) Architectural and tumor cell network ensemble. (**D**) Architectural and tumor cell network with data augmentation. AUC scores were calculated using the five-fold test set in cohort I (*n* = 302).

**Figure 3 cancers-13-03308-f003:**
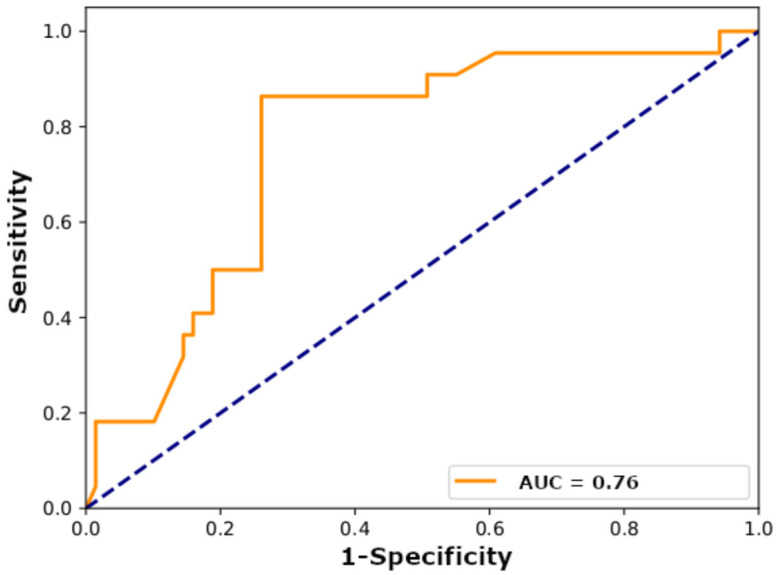
Performance of the DeepRePath model using the architectural and tumor cell network ensemble with data augmentation (100× and 400× magnification pathology images) based on the ROC area under the curve (AUC). AUC scores were calculated in cohort II (*n* = 91).

**Figure 4 cancers-13-03308-f004:**
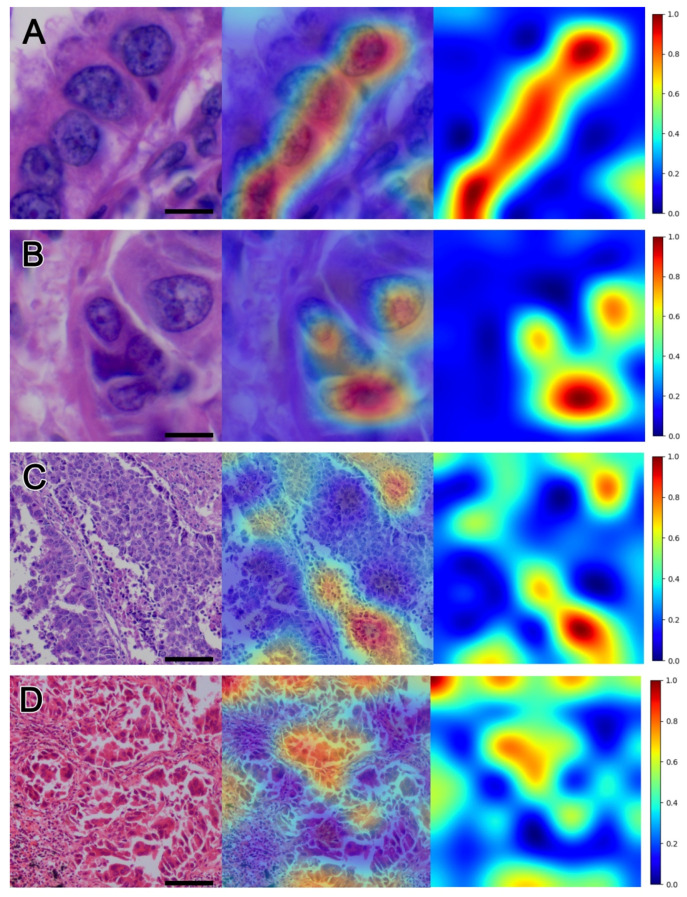
Gradient-weighted class activation mapping (Grad-CAM) heatmaps of pathology images (100× and 400×). Grad-CAM generated heatmaps to identify the histological features that may be associated with a high probability of recurrence. In the high-power images, atypical tumor nuclei (**A**,**B**), tumor necrosis (**C**), and discohesive tumor cells (**D**) were identified using Grad-CAM as associated with a high probability of recurrence. Scale bar length 10 μm (**A**,**B**) and 100 μm (**C**,**D**).

**Figure 5 cancers-13-03308-f005:**
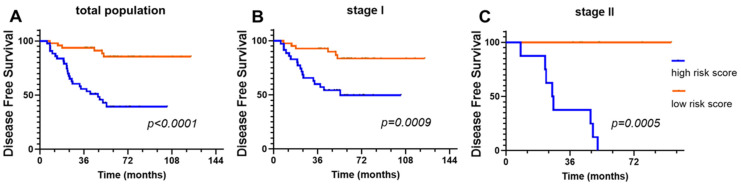
Kaplan–Meier curves according to the predicted risk of recurrence for lung adenocarcinoma patients. The survival of patients with a high-risk score (blue) vs. low-risk score (red) was compared in the total population (**A**), as well as in the stage I (**B**) and stage II (**C**) subgroups in cohort II (external validation set).

**Table 1 cancers-13-03308-t001:** Performance evaluation of cohort I (average of five-fold cross-validation, *n* = 302).

Input Images	Sensitivity (%)	Specificity (%)	PPV (%)	NPV (%)	Accuracy (%)	AUC
Architectural network (100× magnification)	65	59	47	86	62	0.6
Tumor cell network (400× magnification)	52	78	56	84	71	0.68
Architectural + tumor cell ensemble	46	94	78	83	82	0.72
Architectural + tumor cell ensemble (data augmentation)	74	78	59	89	77	0.77

PPV, positive predictive value; NPV, negative predictive value; AUC, area under the curve.

**Table 2 cancers-13-03308-t002:** Performance evaluation of cohort II (external validation set) (*n* = 91).

Input Images	Sensitivity (%)	Specificity (%)	PPV (%)	NPV (%)	Accuracy (%)	AUC
Architectural network (100× magnification)	91	14	25	83	33	0.43
Tumor cell network (400× magnification)	73	62	38	88	65	0.65
Architectural+tumor cell ensemble	59	87	59	87	80	0.75
Architectural+tumor cell ensemble (data augmentation)	86	74	51	94	77	0.76

PPV, positive predictive value; NPV, negative predictive value; AUC, area under the curve.

**Table 3 cancers-13-03308-t003:** Comparison of the performance of models with/without transfer learning for cohort I (training set) and cohort II (external validation set).

Cohort	Model	Sensitivity (%)	Specificity (%)	PPV (%)	NPV (%)	Accuracy (%)	AUC
I	DeepRePath with TL	46	94	78	83	82	0.72
	DeepRePath without TL	80	89	75	93	87	0.87
II	DeepRePath with TL	59	87	59	87	80	0.75
	DeepRePath without TL	64	62	35	84	63	0.58

PPV, positive predictive value; NPV, negative predictive value; AUC, area under the curve; TL, transfer learning.

**Table 4 cancers-13-03308-t004:** Nuclear morphometric results of hotspots and coldspots on patches (400× magnification).

Histopathology	Hotspot (107 Nuclei)	Coldspot (107 Nuclei)	*p*
Area (μm^2^)	16.83 ± 8.39	12.97 ± 5.15	<0.001
Primary axis (μm)	5.59 ± 1.38	4.89 ± 1.06	<0.001
Secondary axis (μm)	3.68 ± 1.04	3.29 ± 0.72	0.001
Maximum Feret (μm)	5.81 ± 1.43	5.09 ± 1.04	<0.001
Minimum Feret (μm)	3.83 ± 1.05	3.39 ± 0.73	<0.001
Perimeter (μm)	15.69 ± 3.73	13.66 ± 2.70	<0.001
Shape factor *	0.821 ± 0.098	0.846 ± 0.070	0.036
Roughness †	0.942 ± 0.037	0.949 ± 0.020	0.101
Aspect ratio ‡	1.584 ± 0.446	1.519 ± 0.326	0.221
Roundness §	0.672 ± 0.155	0.685 ± 0.128	0.526

Values are presented as the mean ± standard deviation. * Shape factor is defined as 4π × area/perimeter^2^. “1” indicates a perfect round nucleus, and as the value approaches “0”, it indicates nuclear shape pleomorphism. † Roughness is defined as area/convex area. “1” indicates minimal irregularity in the nuclear membrane. ‡ Aspect ratio is defined as major axis/minor axis. The higher the value, the longer the nucleus. § Roundness is defined as 4 × area/π × primary axis^2^. A value closer to “1” indicates perfect roundness, and a value closer to “0” indicates an elongated nucleus. *p* < 0.05 was considered statistically significant. All variables were compared using Student’s *t*-test.

**Table 5 cancers-13-03308-t005:** Risk factors for the 3-year recurrence of patients with surgically resected lung adenocarcinoma in the external validation sets (*n* = 91).

Effect	Univariate	*p*	Multivariate	*p*
	Hazard Ratio (95% CI)		Hazard Ratio (95% CI)	
Gender (female vs. male)	0.650 (0.306–1.381)	0.262		
Age (≥60 vs. <60)	1.360 (0.640–2.889)	0.424		
ECOG (≥1 vs. 0)	3.285 (1.604–6.727)	0.001	2.631 (1.267–5.467)	0.009
Tumor grade (moderate to poor vs. well)	2.483 (1.110–5.555)	0.027	1.296 (0.563–2.987)	0.542
Tumor size (≥2.4 vs. <2.4 cm)	1.938 (0.925–4.062)	0.080	2.065 (0.968–4.405)	0.061
LVI (yes vs. no)	2.332 (1.064–5.113)	0.035	1.681 (0.739–3.826)	0.215
pT stage (≥T3 vs. T1–2)	1.490 (0.733–3.030)	0.271		
pN stage (≥N1 vs. N0)	1.763 (0.673–4.621)	0.249		
High vs. low score for recurrence *	6.358 (2.599–15.554)	<0.001	5.564 (2.245–13.789)	<0.001

Gender, female vs. male. Age, ≥60 vs. <60 years. ECOG, Eastern Cooperative Oncology Group. Tumor grade, moderately to poorly differentiated vs. well-differentiated. Tumor size, according to the median value, ≥2.4 vs. <2.4 cm. LVI, lymphovascular invasion, yes vs. no. pT, pathologic T stage, ≥T3 vs. T1–2. pN, pathologic N stage, ≥N1 vs. N0. * the probability scores of DeepRePath model based on the optimal probability threshold using ROC curve for recurrence.

## Data Availability

The program code of DeepRePath and data that do not infringe patients’ personal information are available at https://github.com/deargen/DeepRePath, accessed on 17 May 2021.

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
