# Peer review of "DeepRePath: Identifying the Prognostic Features of Early-Stage Lung Adenocarcinoma Using Multi-Scale Pathology Images and Deep Convolutional Neural Networks"

_cancers, 2021, doi:10.3390/cancers13133308_

Round 1
Reviewer 1 Report
Dear Authors,
I'm happy to read the new version of your work where all points emerged by the first revion step were addressed.
I have only a question about pathologists work of selection of images. The initial slide selection was done by three pathologists (S.A.H., K.Y.,and T-J.K.), but the quality revision of images was performed by only two pathologists (S.A.H. and K.Y.). Why?
Author Response
Dear Reviewers and Editor,
We thank you very much for your consideration of our manuscript, entitled “DeepRePath: identifying the prognostic features of early-stage lung adenocarcinoma using multi-scale pathology images and deep convolutional neural networks” (cancers-1162492), and for the opportunity to resubmit a revised version. We also appreciate the reviewers’ helpful comments on our manuscript. After discussions among all the co-authors, we have revised the manuscript to reflect reviewers’ concerns. We corrected a misspelling in the name of one author (Yeoun Eun Sung). English language and style were rechecked and corrected. Changes in the text are highlighted in yellow. Point-by-point responses to specific reviewer comments are indicated below.
Reviewer 1
- I have only a question about pathologists work of selection of images. The initial slide selection was done by three pathologists (S.A.H., K.Y., and T-J.K.), but the quality revision of images was performed by only two pathologists (S.A.H., and K.Y.). Why ?
Response:
A lot of times and discussions between the pathologists were needed for the quality revision in our study. The pathologist (T-J.K.) did not participate in the work due to the personal schedule. Thus, two pathologists (S.A.H. and K.Y.) conducted the quality revision.
Reviewer 2 Report
The authors addressed the raised issues. Only Table 5 is missing.
Author Response
Dear Reviewers and Editor,
We thank you very much for your consideration of our manuscript, entitled “DeepRePath: identifying the prognostic features of early-stage lung adenocarcinoma using multi-scale pathology images and deep convolutional neural networks” (cancers-1162492), and for the opportunity to resubmit a revised version. We also appreciate the reviewers’ helpful comments on our manuscript. After discussions among all the co-authors, we have revised the manuscript to reflect reviewers’ concerns. We corrected a misspelling in the name of one author (Yeoun Eun Sung). English language and style were rechecked and corrected. Changes in the text are highlighted in yellow. Point-by-point responses to specific reviewer comments are indicated below.
Reviewer 2
- The authors addressed the raised issues. Only Table 5 is missing.
Response:
Thank you for pointing out error. We inserted table 5 in the Result section.

This manuscript is a resubmission of an earlier submission. The following is a list of the peer review reports and author responses from that submission.
Round 1
Reviewer 1 Report
In this study Won Sang Shim and colleagues predicted recurrence using histopathologic images of LUAD. They developed DeepRePath, deep learning model without extracting predefined morphological features for the prognosis prediction of patients with early-stage LUAD. According to authors the model prediction could guide the patient therapy.
I liked the study and approach but currently no sufficient details are described in the text to allow others scientists to replicate and rebuild the showed results.
More attention is needed in the text, “MDPI template instruction for authors” is in the main text (lines 213-228)!!!!
Main points:
1) The study can be considered only a proof of concept for the small number of patients and this aspect must be clarified also in the abstract and conclusion sections.
2) The code/data of the work are not available in a repository (freely or under request). Currently it is not possible replicate and re-build the published results.
3) The code of DeepRePath should be available for results reproducibility.
4) Methods section must be re-written. In the method section I can read MDPI template instruction for authors at line 213-228. Please read the text before re-submission
5) Did you test performance of other boosting techniques or you tested only XGBoost? This aspect could be relevant for the work and it could be addressed.
6) “Pathologists identified a large section of tumor cells on histopathology slides and input the captured images into our deep learning model to determine lung cancer recurrence.” This aspect must be explained in the methods and examples could be showed in supplementary figures. This is relevant to understand the role/effect of pathologist intervention and the power of your model.
7) Three pathologists worked on the input images, but no details are written. Did three pathologists work on each image? More details are needed in the methods section.
8) Statistical analysis is not sufficient detailed in method section and in the figures. It is relevant understanding the statistical test that authors used in each figure and table. Please report statistical test in each legend of table/figure where you used it (For example legend of Table1 is incomplete)
Minor points:
1) Citation of ImageJ software (National Institutes of Health, Bethesda, USA) line 197 and other software that authors used.
2) The size and font of text in supplementary figure 1 should be uniform.
3) At lines 237-238. You report that “The baseline characteristics between cohort I and cohort II were not significantly different except for lymphovascular invasion” but in the table I can read that gender and vascular invasion have p.value < 0.05.
4) A lot of words are reported with “-“. Examples: insuf-ficient in line 131, pa-tients in line 134 rel-atively in line 138 ob-tained in line 194 nu-cleus in line 200 ….
Reviewer 2 Report
See the attached file
